# Antifungal Activity and Mechanism of Xenocoumacin 1, a Natural Product from *Xenorhabdus nematophila* against *Sclerotinia sclerotiorum*

**DOI:** 10.3390/jof10030175

**Published:** 2024-02-26

**Authors:** Shujing Zhang, Yunfei Han, Lanying Wang, Jinhua Han, Zhiqiang Yan, Yong Wang, Yonghong Wang

**Affiliations:** 1Key Laboratory of Plant Protection Resources and Pest Management, Ministry of Education, College of Plant Protection, Northwest A&F University, 22 Xinong Road, Yangling 712100, China; sjzhang@hainanu.edu.cn (S.Z.); hanyunfei@nwafu.edu.cn (Y.H.);; 2Key Laboratory of Green Prevention and Control of Tropical Plant Diseases and Pests, Ministry of Education, School of Tropical Agriculture and Forestry (School of Agricultural and Rural Affairs, School of Rural Revitalization), Hainan University, 58 People’s Avenue, Haikou 570228, China; 3Shaanxi Research Center of Biopesticide Engineering & Technology, College of Plant Protection, Northwest A&F University, 22 Xinong Road, Yangling 712100, China

**Keywords:** *Xenorhabdus nematophila*, Xenocoumacin 1, antifungal activity, *Sclerotinia sclerotiorum*, biopesticide

## Abstract

*Sclerotinia sclerotiorum* (Lib.) de Bary, a polyphagous necrotrophic fungal pathogen, has brought about significant losses in agriculture and floriculture. Until now, the most common method for controlling *S. sclerotiorum* has been the application of fungicides. Xenocoumacin 1 (Xcn1) is a potential biopesticide having versatile antimicrobial activities, generated by *Xenorhabdus nematophila*. This study was intended to isolate Xcn1 from *X. nematophila* YL001 and clarify its efficacies for *S. sclerotiorum* control. Xcn1 demonstrated a wider antifungal spectrum against 10 plant-pathogenic fungi. It also exhibited a strong inhibitory effect on the mycelial growth of *S. sclerotiorum* with an EC_50_ value of 3.00 μg/mL. Pot experiments indicated that Xcn1 effectively inhibited disease extension on oilseed rape and broad bean plants caused by *S. sclerotiorum*. Morphological and ultrastructural observations revealed that the hyphae of *S. sclerotiorum* became twisted, shriveled, and deformed at the growing points after treatment with Xcn1 at 3.00 μg/mL and that the subcellular fractions also became abnormal concurrently, especially the mitochondrial structure. Moreover, Xcn1 also increased cell membrane permeability and decreased the content of exopolysaccharide as well as suppressing the activities of polygalacturonase and cellulase of *S. sclerotiorum*, but exerted no effects on oxalic acid production. This study demonstrated that Xcn1 has great potential to be developed as a new biopesticide for the control of *S. sclerotiorum*.

## 1. Introduction

*Sclerotinia sclerotiorum* (Lib.) de Bary is a polyphagous and necrotrophic fungal pathogen which infects more than 400 plant species worldwide, such as oilseed rape [1], lentils [2], sunflower [3], soybean [4], chickpeas [5], lentils [6], and lettuce [7], resulting in significant losses in agriculture and floriculture. *S. sclerotiorum* infects different plant tissues and produces sclerotia, which are the source of infection in the disease cycle. Moreover, sclerotia are structures formed from converged mycelia and can survive in the soil for up to 8 years [8,9]. Oxalic acid (OA), which inhibits plant defense, is a key pathogenicity factor of *S. sclerotiorum* [10]. Effectively eradicating *S. sclerotiorum* requires preventing the production of oxalic acid and the formation of sclerotia. Owing to the broad host range and the persistent sclerotia bodies of *S. sclerotiorum*, as well as its ability to infect several rotational crops, traditional strategies including crop rotation and breeding of resistant crops are not effective in controlling *S. sclerotiorum* [11]. Currently, the most common method of *S. sclerotiorum* disease management is the application of chemical fungicides [12]. However, their extensive use makes sclerotia generate drug resistance to many fungicides, such as mancozeb, propineb, and carbendazim [13,14]. In addition, the adverse side effects of these chemical fungicides have also resulted in serious threats to human health and environmental security [15,16,17].

*Xenorhabdus nematophila*, which is a form of entomopathogenic bacteria, lives in symbiosis with *Steinernema* nematodes [18]. Previous studies have shown that cell-free broth derived from *X. nematophila* has antifungal activity, but it is unknown which compounds have antifungal effects [19]. Xenocoumacin 1 (Xcn1), a type of xenocoumacin, is the main antimicrobial compound produced by *X. nematophila* [20]. A previous study has shown that Xcn1 exhibits a strong antimicrobial activity against five species of *Phytophthora*, with EC_50_ values ranging from 0.25 to 4.17 µg/mL [21]. The EC_50_ values of Xcn1 against mycelial growth and zoospore germination of *Phytophthora capsici* were 2.44 and 0.81 µg/mL, respectively [22]. The transcriptional responses of *Bacillus subtilis* to Xcn1 predicted that its functions were similar to the mechanisms of protein synthesis inhibitors [23]. However, information regarding Xcn1 relating to its efficacy against *S. sclerotiorum* is limited.

It is well known that living microbes and their metabolites are important resources for the development of new biopesticides. Thus, identifying new compounds from microbes and evaluating their antimicrobial activity and action mechanism are important. Our previous studies revealed that the culture supernatant of *X. nematophila* YL001 exhibited potent antimicrobial activities against many plant pathogens [24,25,26]. However, to date, few bioactive compounds have been obtained from *X. nematophila* YL001. We have obtained Xcn1 from cell-free broth derived from *X. nematophila* YL001. Therefore, the objectives of the present study were to (i) evaluate inhibitory effects of Xcn1 against plant pathogens in vivo and in vitro; (ii) verify the effects of Xcn1 on the hyphal morphology and ultrastructure of *S. sclerotiorum*; (iii) test the effects of Xcn1 on the cell membrane permeability and the exopolysaccharide (EPS) and oxalic acid (OA) production of *S. sclerotiorum*; (iv) determine the effects of Xcn1 on the activities of polygalacturonase (PG) and cellulose (Cx) of *S. sclerotiorum*. The above results provided basis information for the potential of Xcn1 as a new candidate for biopesticides, as well as its action mechanism against *S. sclerotiorum*.

## 2. Results

### 2.1. Structure Elucidation and Antifungal Activity Assay of Xcn1

Xcn1 was isolated and purified from cell-free fermentation broth derived from *X. nematophila* YL001 using X-5 macroreticular resin, cation-exchange resin 110, and CM-Sephadex-C-25 weak acid cation exchange resin consecutively. Its structure was identified correctly via ^1^H NMR, ^13^C NMR, and HR-MS spectra (Appendix A) [20,21]. Its purity was determined to be 96.7% via the HPLC method (Appendix A).

A previous study described the satisfactory antimicrobial activity of Xcn1 against a species of *Phytophthora* [21]. On the basis of this research, we further determined Xcn1’s antifungal activity against ten plant-pathogenic fungi. Of the fungal pathogens tested, Xcn1 showed high inhibitory effects on *S. sclerotiorum*, *R. solani*, *B. cinerea*, and *E. turcicum* at 10 µg/mL, with inhibition rates of 89.30%, 88.5%, 86.7% and 83.2%, respectively (Table 1). *S. sclerotiorum* was the most sensitive to Xcn1 with an EC_50_ value of 2.86 µg/mL (Table 2), indicating the strong antifungal activity of Xcn1 against *S. sclerotiorum*.

### 2.2. Effects of Xcn1 on the Disease Development In Vivo

The disease control efficacies of Xcn1 against *S. sclerotiorum* in two host plants (broad bean and oilseed rape) are shown in Figure 1 and Figure 2, respectively. As shown in Figure 1 and Appendix A, Xcn1 effectively protected the broad bean leaves from *S. sclerotiorum* infection, with preventive efficacies of 62.27% at 100 µg/mL and 78.99% at 200 µg/mL (Figure 1A,C and Appendix A). Moreover, Xcn1 also exhibited high curative activity against *S. sclerotiorum* on the infected diseased leaves, with control efficacies of 66.51% at 100 µg/mL and 88.90% at 200 µg/mL, which is comparable to that of carbendazim (MBC) treatment at 100 µg/mL (92.01%) (Figure 1B,D and Appendix A).

The control efficacy of Xcn1 against *S. sclerotiorum* in the oilseed rape plants is shown in Figure 2 and Appendix A. Xcn1 possessed a favorable preventive efficacy, with a control efficacy of 58.40% at 100 µg/mL and an efficacy of 86.30% at 200 µg/mL, comparable to that of carbendazim (MBC) treatment at 100 µg/mL (91.30%) (Figure 2A,C and Appendix A). Xcn1 also exhibited an excellent curative activity, with a control efficacy of 45.79% at 100 µg/mL, comparable to that of the referenced carbendazim (MBC) at 100 µg/mL (48.92%) (Figure 2B,D and Appendix A). Taken together, the disease control assay indicated that Xcn1 exhibited a potent control efficacy against *S. sclerotiorum* in vivo.

### 2.3. Effects of Xcn1 on the Hyphal Morphology and Ultrastructure of S. sclerotiorum

Previous research has reported that Xcn1 brings about the plasmolysis and cytoplasmic condensation of *P. capsici* [22]. To better understand the mode of antifungal action of Xcn1, we observed the mycelial morphology and ultrastructure of *S. sclerotiorum* treated with Xcn1 via SEM and TEM, respectively. The SEM results showed that the control sample exhibited a fine morphology of smooth, uniform, and cylindrical hyphae with plump growing points (Figure 3A,B). The Xcn1-treated sample, however, showed an altered morphology characterized by twisted, deformed, and rupturing hyphae with a sparse distribution (Figure 3C–F).

The effects of Xcn1 on the hyphal ultrastructure of *S. sclerotiorum* were observed via TEM. The control sample displayed a normal hyphal ultrastructure of intact and smooth cell membranes, as well as clearly visible protoplasm and mitochondria (Figure 4A,B). In the sample treated with Xcn1, the hyphae’s cell membranes became rough and invaginated. Mitochondrial abnormalities, including hazy outlines and vacuolar degeneration, were also observed. Moreover, the cytoplasm became disordered and vacuolate (Figure 4C–F). These microscopic observations indicated that Xcn1 disrupted the hyphal morphology and ultrastructure of *S. sclerotiorum* to exert its inhibitory effect on mycelium growth.

### 2.4. Effects of Xcn1 on the Cell Membrane Permeability and Exopolysaccharide Secretion of S. sclerotiorum

As shown in Figure 5A, relative conductivity increased over time whether the strains were treated with Xcn1 or not. The relative conductivity of the Xcn1-treated group (3.00 μg/mL), however, was always significantly higher than that of the control group (Figure 5A). These results indicated that Xcn1 increased the cell membrane permeability of *S. sclerotiorum*.

The effects of Xcn1 on the EPS secretion of *S. sclerotiorum* were determined to be due to the phenol-sulfuric acid method and were reported as glucose equivalents. As illustrated in Figure 5B, there were no differences in the content of EPSs between the Xcn1-treated group and the control group at the 3th day after inoculation. However, the EPS contents of the Xcn1-treated group were lower than those of the control group at the 6th, 9th and 12th days, respectively (Figure 5B). Visually, the supernatant culture of *S. sclerotiorum* after treatment with Xcn1 for 9 days was more limpid than that of the control group (Appendix A).

### 2.5. Effects of Xcn1 on the Oxalic Acid Biosynthesis of S. sclerotiorum

Bromophenol blue is an acid–base indicator, which exhibits a yellow color at low pH values below 3.0 and a blue color at pH values above 4.6 [27]. *S. sclerotiorum* produced OA to acidize the medium (pH < 3.0) during its growth [28]. Thus, we evaluated the effect of Xcn1 on this acidification process on the PDA medium supplemented with bromophenol blue. As shown in Figure 5C, although the growth of *S. sclerotiorum* was obviously suppressed by Xcn1 compared with that of the control, the acidification process still occurred on the Xcn1-treated plates, albeit with a time delay. On the basis of this result, we further tested the effects of Xcn1 on the OA production of *S. sclerotiorum*. As shown in Figure 5D, there were no significant differences in the content of OA between the Xcn1-treated sample and the control sample on a timeline extending from 3 to 12 days. All these results reveal an interesting idea, namely that Xcn1 has no effects on the biosynthesis of OA in *S. sclerotiorum*.

### 2.6. Effects of Xcn1 on the Activities of Polygalacturonase and Cellulase in S. sclerotiorum

The PG and Cx activities of *S. sclerotiorum* treated with Xcn1 were measured with the results shown in Figure 5. The PG activities of the Xcn1-treated group were comparable to those of the control group at the 3rd and 6th days after inoculation. However, when the inoculation time extended to the 9th and 12th days, the PG activities of the Xcn1-treated group were much lower than those of the control group (Figure 5E). Moreover, the Cx activities of the Xcn1-treated group were significantly lower than those of the control group at the 6th, 9th and 12th days after inoculation, respectively (Figure 5F). All these results indicated that Xcn1 suppresses the activities of PG and Cx in *S. sclerotiorum* when the interaction time is over 6 days.

## 3. Discussion

### 3.1. Xcn1 Has the Potential to Be Developed as a Pesticide for the Control of S. sclerotiorum

Xenocoumacin 1 (Xcn1) has great potential to be used as a new biopesticide in agricultural production. A previous study has reported that Xcn1 displayed good antimicrobial activity against *Phytophthora* species, especially against *P. capsici*, with an EC_50_ value of 2.44 μg/mL [21]. In this study, Xcn1 was isolated and purified from *X. nematophila* YL001. To obtain more information about its pesticidal properties, we tested its antifungal activity in vitro and in vivo. Xcn1 exhibited broad-spectrum antimicrobial activity against 10 common phytopathogens in vitro (Table 1) and was especially effective against *S. sclerotiorum*, with an EC_50_ value of 3.0 µg/mL. Moreover, Xcn1 was demonstrated to possess a comparable in vivo control efficacy against *S. sclerotiorum* in two selected host plants at 200 µg/mL to MBC at 100 µg/mL (Figure 1 and Figure 2). These results indicate that Xcn1 has the potential to be developed as a microbiological pesticide for the control of *P. capsici*, *S. sclerotiorum*, and other pathogens.

### 3.2. Xcn1 Exerted Its Antifungal Activity against S. sclerotiorum through Disturbing Mycelial Growth and Infection

Xcn1 exerted its antifungal activity against *S. sclerotiorum* by disturbing the multiple physiological and biochemical processes. SEM and TEM observations revealed prominent morphological and ultrastructural disruptions of *S. sclerotiorum* hyphae treated with Xcn1 compared to the untreated controls (Figure 3 and Figure 4). Xcn1 increased the cell membrane permeability of *S. sclerotiorum* (Figure 5A), as well as reducing the secretion of EPSs (Figure 5B and Appendix A), an important virulence factor of many kinds of microorganism [29,30,31]. Plant pathogens produced such cell wall-degrading enzymes as cellulase, hemicellulase, and pectinase to break down the first barrier of the host plant cells during their invasion. Pectinase and cellulase have been demonstrated to play key roles during the infection of *S. sclerotiorum* [32,33,34]. In this study, we also found that Xcn1 suppresses the activities of PG and Cx in *S. sclerotiorum* effectively (Figure 5). However, more work is needed to further clarify its mechanism of action as a potent antifungal agent.

### 3.3. Xcn1 May Inhibit Protein Translation

Amicoumacin A (AMI), a close analog of Xcn1, has been shown to bind to the E-site of the ribosome [35]. A published study has indicated Xcn1 could be an inhibitor of ribosomal protein synthesis; however, there is no direct experimental evidence to support this viewpoint [36]. Xcn1 has a longer carbon chain than AMI and contains guanidine groups that AMI does not have. Therefore, it cannot be confirmed that the mechanisms of action of AMI and Xcn1 are the same. Blasticidin S has a long chain structure containing guanidine groups similar to Xcn1 and inhibits the hydrolysis of tRNA-carrying peptides, resulting in the inability to release synthesized peptides and cause translation termination [37,38]. These studies suggest that the long carbon chain and guanidine structure of Xcn1 may play an important role in inhibiting protein translation processes.

### 3.4. Combined Utilization of Xcn1 and Oxalate Synthesis Inhibitors Will Be a High Effective Strategy in S. sclerotiorum Control

Combined utilization of Xcn1 and oxalate synthesis inhibitors may achieve the effective control of *S. sclerotiorum*. The present knowledge points out that oxalic acid (OA) is a key factor in successful infection by *S. sclerotiorum* [39]. OA is an elicitor of programmed plant cell death in the process of *S. sclerotiorum* infection, and its deficiency reduces the pathogenicity of *S. sclerotiorum* [40]. However, Xcn1 has been demonstrated to have no effects on the production of OA in *S. sclerotiorum* (Figure 5C,D). Many fungicides, such as pyraziflumid [41], pyrisoxazole [1], cinnamic acid [42], volatile organic compounds from *Bacillus* spp. [11], and lansiumamide B [43], have been reported to reduce the content of OA to weaken the pathogenicity of *S. sclerotiorum*. Combination of these compounds with Xcn1 may be a highly effective control strategy using multiple modes of action.

## 4. Material and Methods

### 4.1. Strains and Growth Conditions

*X. nematophila* YL001 was obtained from its host nematodes, *Steinernema* sp. YL001, which were isolated from the soil of Yangling, China. Its morphological and molecular characteristics were identified [26,44]. *X. nematophila* YL001 was shaken in Luria–Bertani medium (LB: 1.0% Bacto tryptone, 0.5% yeast extract and 1% NaCl in water; pH 7.2) at 150 rpm at 28 °C and transferred to NBTA medium for incubation at 28 °C in darkness [26].

Ten plant-pathogenic fungi for testing (*Sclerotinia sclerotiorum*, *Rhizoctonia solani*, *Exserohilum turcicum*, *Fusarium graminearum*, *Verticillium dahliae*, *Botrytis cinerea*, *Alternaria alternata*, *Colletotrichum gloeosporioides*, *Gaeumannomyces graminis* and *Fusarium oxysporum*) were provided by the Shaanxi Research Center of Biopesticide Engineering & Technology. Their detailed information is presented in Appendix A.

### 4.2. Microbial Fermentation

*X. nematophila* YL001 was inoculated in a 500 mL flask containing 200 mL fresh Luria–Bertani medium and cultured for 12 h at 28 °C while being shaken at 180 rpm. Then five liters of culture broth (OD_600_ = 0.8) was transferred as a seed into a 70-L fermenter (Eastbio, Zhenjiang, China) containing 35 L of TSB medium [25]. The system was cultured under identical conditions (28 °C, pH 7.2, agitation of 160 rpm/min, and 0.25 *v*/*v*/min of aeration) for 48 h. After that, *X. nematophila* YL001 cell-free culture broth was obtained via centrifugation at 10,000× *g* for 20 min. Isolation and characterization of Xcn1 are shown in the Appendix A [20,45].

### 4.3. Assessment of Antifungal Activity of Xcn1 In Vitro

The antifungal activity of Xcn1 against the mycelial growth of phytopathogenic fungi was performed as described, with a slight modification [26]. Briefly, Xcn1 was dissolved in distilled water to prepare a stock solution of 1000 μg/mL, and carbendazim (MBC) was dissolved in 0.1 M HCl with a concentration of 500 μg/mL. The stock solution of Xcn1 (150 μL) was sufficiently mixed with 15 mL of molten PDA medium at a low temperature (<40 °C), and then the mixture was poured into Petri dishes (60-mm diameter) to form plates of a 2–3 mm thickness (5 mL for each disc). The final concentration of Xcn1 in each plate was 10 μg/mL. A 5-mm dish of actively growing (2-day-old) culture of fungus was transferred from the edges of 2-day-old colonies to the center of the prepared plates (one for each plate). PDA plates supplemented with MBC (0.5 μg/mL) served as the positive controls. Mean growth values were measured and subsequently converted into an inhibition rate of mycelial growth in relation to the control treatment according to the following formula:Inhibition rate (%) = [(Dc − D_t_)/(D_c_ − 0.5)] × 100 
where D_c_ and D_t_ represent the mycelial growth diameters (cm) of the control and the treatment group, respectively; 0.5 represents the diameter of mycelial plugs (cm).

The effective concentration for 50% inhibition (EC_50_) was derived from the data analysis of the concentration inhibition rate. Serial concentrations of Xcn1 solutions were separately mixed with the molten PDA medium to the final concentrations of 0.1, 0.5, 1, 2, 4, 6, 8 and 10 μg/mL, and were poured into Petri dishes (60 mm diameter) to form plates. After inoculation of a 5-mm fungal mycelia disc into the center of the solidified medium, the dishes were incubated in the dark at 25 °C for 2 days. The inhibition rate was calculated via the above method when the control fungal mycelium reached the edge of the dishes. All of experiments were independently performed three times under the same conditions.

### 4.4. Assessment of Antifungal Activity of Xcn1 In Vivo

The efficacy of Xcn1 on broad bean and rape seedlings infected with *S. sclerotiorum* was evaluated according to a previous study with some modifications [46]. Seedlings of rape and broad bean plants were grown in a growth chamber (25 ± 1 °C, 75 ± 10% RH, and a 12:12 LD photoperiod) for 25 days. Stocks of Xcn1 (100 and 200 µg/mL) were prepared in sterile water composed of 0.5% Tween 20 (*v*/*v*). All experiments were arranged in a plant growth chamber with three plants per treatment and repeated three times.

For the protective activity assay, rape seedlings and broad bean leaves were sprayed with water, MBC (100 µg/mL), and Xcn1 (100 and 200 µg/mL) separately using a manually operated mini-knapsack sprayer (5 mL for each plant) [42]. The sprayer was held 20 cm away from the plant. After incubation for 12 h, each leaf was wounded by a sterilize needle at the center and inoculated with a 5-mm mycelia agar disc. Finally, the plants were placed in a plant growth chamber. Images were captured after incubation at 25 °C for two days. The lesion area was quantified using Image J 1.38× (https://imagej.nih.gov/ij/ accessed on 18 March 2019), and the control efficacy was calculated according to the following formula:Control efficacy (%) = [(A_c_ − A_t_)/A_c_ − 0.25] × 100 
where A_c_ and A_t_ represent the disease areas of the control group and the treatment group, respectively; 0.25 represents the area of the mycelial agar disc (cm^2^).

For the curative activity assay of agentia to the broad bean and rape seedlings a 5-mm mycelia agar disc was placed in the center of the leaves. The plants were incubated for 12 h (~3 mm lesion around the discs), and then they were sprayed with the drugs as described above. The following procedures were same as those used for the protective activity assay.

### 4.5. Scanning Electron Microscopy (SEM) and Transmission Electron Microscopy (TEM) Observations

The variations of hyphal morphology and ultrastructure of *S. sclerotiorum* treated with Xcn1 was investigated according to the method used by Soner Soylu [47]. Mycelial agar discs that had been growing for two days were placed in the middle of PDA plates with 3.0 μg/mL (EC_50_ value) of Xcn1 (one disc for each plate) and incubated at 25 °C for 48 h in darkness. PDA plates without Xcn1 were used as controls.

A 2.5% glutaraldehyde fixation of mycelial discs (2 mm × 4 mm × 4 mm) in 0.1 M phosphate buffer (pH = 7.2) was conducted overnight for scanning electron microscopy (SEM) observations. Next, with an identical buffer for each disc, they were washed three times for 20 min to remove the glutaraldehyde. The fixed specimens were dehydrated in a gradually increasing concentration of ethanol series (twice at 30%, 50%, 70%, 80%, and 90%, respectively, and three times at 100%) for 20 min for each succession. After dehydration, the discs were dipped into isoamyl acetate for replacement three times for 20 min each. Finally, each specimen was dried using supercritical carbon dioxide. With double-sided tape, the dried samples were fastened to a sample stage and given a gold coating using an E1010 sputter-coating machine (Hitachi, Tokyo, Japan) for 90 s at 9 mA. The images were recorded using a JSM-6360LV SEM (JEOL, Tokyo, Japan).

For transmission electron microscopy (TEM) observations, the pre-fixed mycelial discs were prepared as described in the SEM method. The discs were fixed with 1% osmic acid for 2 h after washing. Then, the specimens were washed again, immediately followed by dehydration as previously described. Samples were dipped into ethanol twice (30 min each), after which the specimens were passed through a solution of epoxy resin/epoxy propane (1:1, *v*/*v*) for 1 h and embedded in epoxy medium at 55 °C for 48 h. Subsequently, blocks were cut into slices using a diamond knife with an Ultramicrotome blade (Leica-ULTRACUT, Wetzlar, Germany) into ultrathin sections of approximately 70 nm. The ultrathin sections were contrasted with 2% uranyl acetate and 2% lead citrate for 30 min prior to examination on a JEM-1230 TEM (Hitachi, Tokyo, Japan).

### 4.6. Determination of Oxalic Acid (OA) Production of S. sclerotiorum

The medium acidification assay was verified on bromophenol blue agar plates with some modifications [11]. Molten PDA medium (30 mL) with 50 µg/mL of bromophenol blue was sufficiently mixed with Xcn1 at a low temperature (<40 °C) to the final concentration of 3.0 µg/mL, and the mixture was poured into Petri dishes (90-mm diameter) to form plates of a thickness of 2–3 mm (10 mL for each dishes). Mycelial plugs (5 mm in diameter) were transferred from the edges of 2-day-old colonies to the center of the plates (one plug for each plate). The plates without Xcn1 served as the controls. The Petri dishes were incubated at 25 °C in darkness, and the pH change was observed at 24 h intervals as the color shifted from blue to yellow.

Oxalic acid released in PDB medium by *S. sclerotiorum* isolates was quantified using the protocol of a previous study, with minor modifications [48]. Briefly, ten agar plugs of *S. sclerotiorum* were grown in a 250 mL Erlenmeyer flask containing 100 mL fresh PDB medium. The flasks were incubated on a rotary shaker at 25 °C with 150 rpm. After 24 h, Xcn1 was added to the flasks, with the final concentration being 3.0 μg/mL. Flasks without Xcn1 were used as the controls. The flasks were then statically incubated over different time periods (3, 6, 9, and 12 d). After that, the filtrates were centrifuged at 5000× *g* for 20 min, and the supernatants were used for oxalic acid determination. The weight of the mycelia of each sample was determined after lyophilization. The oxalic acid concentration was calibrated using the dry mycelium weight of each sample.

### 4.7. Measurement of the Cell Membrane Permeability of S. sclerotiorum

For the measurement of cell membrane permeability, *S. sclerotiorum* colonies were processed according to the method described in [42]. Ten agar plugs of *S. sclerotiorum* isolates were grown in a 250 mL Erlenmeyer flask containing 100 mL PDB. After 24 h, Xcn1 was added to the flasks, with the final concentration being 3.0 μg/mL. Flasks without Xcn1 were used as the controls. The flasks were then incubated at 25 °C and shaken at 150 rpm. After the flasks were shaken for 72 h, the mycelia were collected and washed three times with sterilized water. Then 1 g of fresh mycelia per sample was suspended in 50 mL of distilled water. The conductivity of the system was measured at 0, 1, 2, 4, 6, 8 and 10 h intervals via a conductivity meter (CON510 Eutech/Oakton, Singapore). After that, the samples were boiled for 10 min to measure final conductivity. Each treatment had three replicates, and the experiment was performed twice. The relative conductivity of mycelia was calculated as follows:Relative conductivity = C_d_/C_f_ × 100%

C_d_ represents the conductivity at a certain time interval, C_f_ represents the final conductivity.

### 4.8. Determination of the Exopolysaccharides of S. sclerotiorum

The effects of Xcn1 on exopolysaccharides of *S. sclerotiorum* were determined via the phenol-sulfuric acid method [48,49]. To establish the standard curve of EPSs, 2 mL of different glucose solutions (0, 20, 40, 60, 80, 100, 120, 160 and 200 µg/mL) were added to the different tubes. Then 1 mL of 5% phenol solution was added to each of the above tubes, and the solutions in each tube were mixed gently. Finally, 5 mL of H_2_SO_4_ was added to each tube slowly. The solutions in each tube were mixed gently for 10 s, and then incubated at 25 °C for 30 min. A standard curve was generated by plotting the absorbance at 490 nm against the glucose concentration.

For determination of EPS, the sample supernatants were prepared as described in Section 4.7. EPS was precipitated from 10 mL of each supernatant with 30 mL of absolute ethanol and dried at 50 °C. The sediment of EPS was dissolved in 10 mL of distilled water and quantified by the method of the standard curve. Sterile-distilled water was used as a control. Each treatment was set three replications, and the test was repeated three times.

### 4.9. Measurement of Activities of Polygalacturonase (PG) and Cellulase (Cx) of S. sclerotiorum

The activities of PG and Cx of *S. sclerotiorum* were measured according to the method of a previous study, with minor modifications [39]. Ten mycelial discs (5 mm diameter) were placed in a 250 mL Erlenmeyer flask containing 100 mL of Czapek’s liquid medium (NH_4_NO_3_ 0.2%, KH_2_PO_4_ 0.1%, MgSO_4_ 0.01%, yeast extract 0.05%, NaOH 0.1%, vitamin B_1_, 3 g/L; pH 5.0) with 1% citrus pectin (Sigma Chemical, grade I) as the carbon source [50]. After 24 h, Xcn1 was added to the flasks, with the final concentration being 3.0 μg/mL. Flasks without Xcn1 were used as the controls. Flasks were incubated on a rotary shaker at 25 °C and shaken at 150 rpm. Culture filtrates were collected at the fixed time periods (3, 6, 9, and 12 d). The filtrates were centrifuged at 8000× *g* for 20 min, and the supernatants were stored at −80 °C as the crude enzyme solutions for use.

### 4.10. Data Analysis

Multiple comparisons between treatments were evaluated according to the least significant differences (l.s.d) at *p* < 0.05. The EC_50_ values were calculated statistically by probit analysis using the probit package of SPSS 23.0 software (SPSS Inc., Chicago, IL, USA). The lesion areas were quantified from the captured images with an Image J 1.38× software (https://imagej.nih.gov/ij/, accessed on 18 March 2019).

## 5. Conclusions

Xcn1 exhibited a wider antifungal spectrum against 10 plant-pathogenic fungi and was most effective against *S. sclerotiorum* with excellent antifungal activities in vitro and in vivo. Xcn1 disturbed the mycelial growth and infection of *S. sclerotiorum* and has great potential to be developed as a new biopesticide for disease control.

## Figures and Tables

**Figure 1 jof-10-00175-f001:**
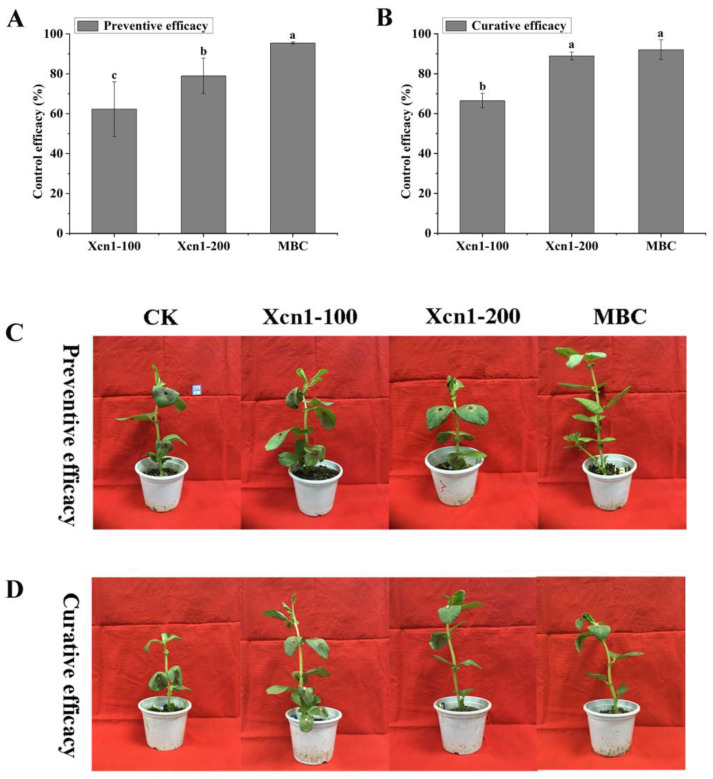
In vivo antifungal activity of Xcn1 and carbendazim against *S. sclerotiorum* in broad bean. (**A**,**C**) Protective activity of Xcn1. (**B**,**D**) Curative activity of Xcn1. Preventive/curative efficacy: a sample solution was sprayed on broad bean leaves 12 h before/after inoculation with *S. sclerotiorum*. Xcn1-100, 100 µg/mL of Xcn1; Xcn1-200, 200 µg/mL of Xcn1; MBC: 100 µg/mL of carbendazim. The different lowercase letters in the bar chart indicate significant differences according to Fisher’s prtected (l.s.d) test (*p* = 0.05).

**Figure 2 jof-10-00175-f002:**
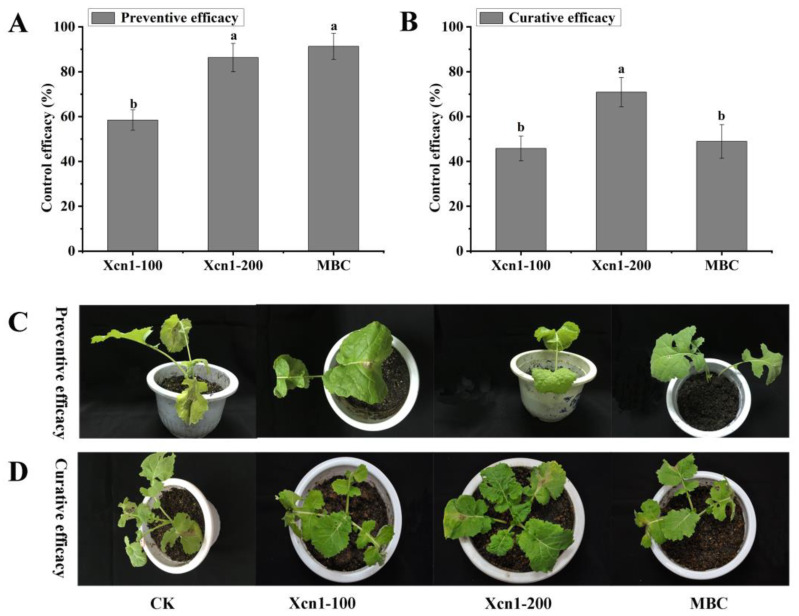
In vivo antifungal activity of Xcn1 and carbendazim against *S. sclerotiorum* in rape seedlings. (**A**,**C**) Protective activity of Xcn1. (**B**,**D**) Curative activity of Xcn1. Preventive/curative efficacy: a sample solution was sprayed on broad bean leaves 12 h before/after inoculation with *S. sclerotiorum*. Xcn1-100, 100 µg/mL of Xcn1; Xcn1-200, 200 µg/mL of Xcn1; MBC: 100 µg/mL of carbendazim. The different lowercase letters in the bar chart indicate significant differences according to Fisher’s prtected (l.s.d) test (*p* = 0.05).

**Figure 3 jof-10-00175-f003:**
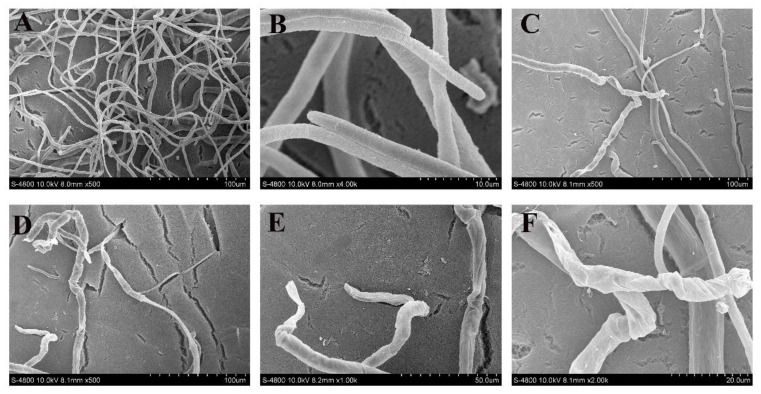
Effects of Xcn1 on the hyphal morphology of *S. sclerotiorum*. (**A**,**B**) Healthy hyphae in control plates. (**C**–**F**) Hyphae treated with Xcn1 (3.0 µg/mL). Scale bar: 100 µm for (**A**,**C**,**D**); 10 µm for B; 50 µm for (**E**); 20 µm for (**F**).

**Figure 4 jof-10-00175-f004:**
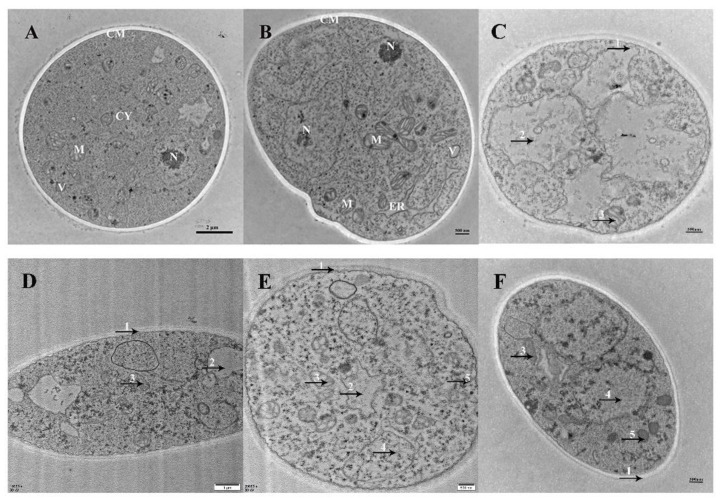
Effects of Xcn1 on the hyphal cell ultrastructure of *S. sclerotiorum*. (**A**,**B**) Healthy hyphae in control plates. (**C**–**F**) Hyphae treated with Xcn1 (3.0 µg/mL). Xcn1: plates treated with Xcn1 at 3.0 µg/mL; 1, the invaginated cell membrane; 2 and 4, vacuolation degradation; 3, mitochondrial abnormalities; 5, cytoplasmic disorders; M: mitochondria, ER: endoplasmic reticulum, N: nucleus, V: vacuole, CY: cytoplasm, and CM: cell membrane. Scale bar: 2 µm for (**A**); 500 nm for (**B**,**C**,**E**,**F**); 1 µm for (**D**).

**Figure 5 jof-10-00175-f005:**
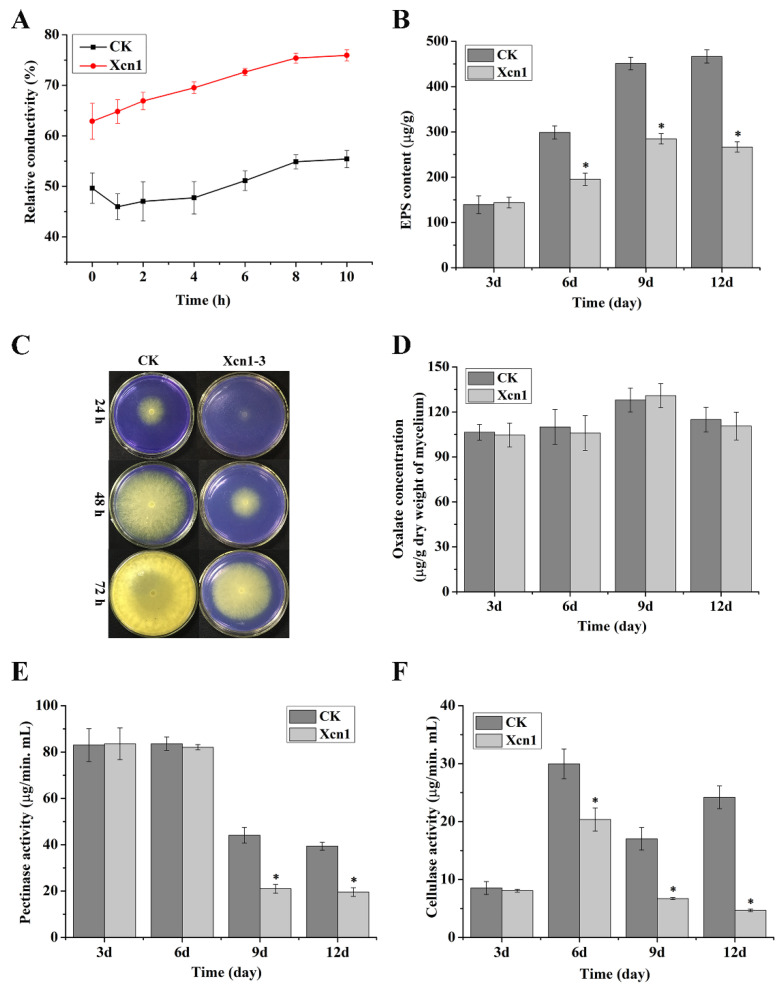
Pharmacological characteristics of Xcn1 on the *S. sclerotiorum*. (**A**) The relative conductivity variations of the mycelia of *S. sclerotiorum* treated with Xcn1 (3.0 μg/mL) or not (CK) for 72 h. (**B**) The exopolysaccharide (EPS) secretion contents of *S. sclerotiorum* treated with Xcn1 (3.0 μg/mL) or not (CK) at different time intervals (3, 6, 9, and 12 d). (**C**) Medium acidification assay of *S. sclerotiorum* treated with Xcn1 (3.0 μg/mL) or not (CK) at different time intervals (1, 2, and 3 d). (**D**) The OA contents of *S. sclerotiorum* treated with Xcn1 (3.0 μg/mL) or not (CK) at different time intervals (3, 6, 9, and 12 d). (**E**) Polygalacturonase (PG) activities of *S. sclerotiorum* treated with Xcn1 (3.0 μg/mL) or not (CK) at different time intervals (3, 6, 9, and 12 d). (**F**) Cellulase (Cx) activities of *S. sclerotiorum* treated with Xcn1 (3.0 μg/mL) or not (CK) at different time intervals (3, 6, 9, and 12 d). Statistical differences were detected between Xcn1 treatment and the CK with independent sample t tests (asterisk [*] indicates *p* < 0.05).

**Table 1 jof-10-00175-t001:** The inhibition rates of Xcn1 against 10 plant-pathogenic fungi.

Tested Fungi	Inhibition Rate (%)	
	Xcn1 ^a^	MBC ^b^
*Sclerotinia sclerotiorum*	89.30 ± 2.30	95.17 ± 4.05
*Rhizoctonia solani*	88.50 ± 3.10	67.70 ± 3.15
*Botrytis cinerea*	86.70 ± 3.20	13.88 ± 2.28
*Exserohilum turcicum*	83.20 ± 2.20	-
*Alternaria alternata*	75.00 ± 1.80	-
*Fusarium graminearum*	50.60 ± 1.29	89.78 ± 3.08
*Alternaria solani*	67.50 ± 2.10	65.58 ± 3.52
*Colletotrichum gloeosporioides*	47.01 ± 3.10	56.80 ± 2.24
*Gaeumannomyces graminis*	38.11 ± 3.40	44.05 ± 2.12
*Fusarium oxysporum*	11.80 ± 2.20	34.54 ± 3.87

^a^ The concentration of Xcn1 was set at 10 µg/mL. ^b^ The concentration of carbendazim (MBC) was set at 0.5 µg/mL. Data are represented as the mean of three replications ± standard deviation.

**Table 2 jof-10-00175-t002:** Antifungal activity of Xcn1 against *S. sclerotiorum*.

Strain	Regression Curve	EC_50_ ^a^ (CI_95_ ^b^) (μg/mL)	EC_95_ ^c^ (CI_95_ ^b^) (μg/mL)	Chi ^d^
*S. sclerotiorum*	y = 2.01x − 0.92	2.86 (2.38–3.32)	18.71 (14.49–26.65)	2.21

^a^ Effective dose for 50% inhibition compared with the control. ^b^ 95% confidence intervals. ^c^ Effective dose for 95% inhibition compared with the control. ^d^ Chi-square value, significant at *p* < 0.05.

## Data Availability

The original contributions presented in the study are included in the article/Appendix A, further inquiries can be directed to the corresponding author.

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
