# Peer review of "Antifungal Activity and Mechanism of Xenocoumacin 1, a Natural Product from Xenorhabdus nematophila against Sclerotinia sclerotiorum"

_jof, 2024, doi:10.3390/jof10030175_

Round 1

Reviewer 1 Report

Comments and Suggestions for Authors

The work is interesting, so I congratulate the authors on the making of the work. I do have some observations, that are explained in the following paragraphs.

In the introduction, the authors must explain in a more detailed manner the problems with erradication of S. sclerotiorum, the importance of oxalic acid production by S. sclerotiorum, briefly infection strategies used by S. sclerotiorum, for example, to better undestand why of the experiments. Also, I would like the authors to include more details on the compund Xcn1, for example, which other microorganisms produce such compound or similar ones, does all other similar compounds have similar effects to Xcn1, are there any proposed mechanism of action for xenocoumacins in general.

In the Material and Methods section, please provide a reference for the used concentrations of MBC antifungal, or justify why use such concentration. Also, please check for the correct use of italics on the scientifc names of the organisms used in this work, as well as the correct names of: Verticillium dahliae (the las "e" is missing), Botrytis cinerea (remove the "l" from "cinereal" ), Alternaria alternata (change the las "e" on "alternate" to "a"), in lines 256 and 257.

One of the main problems with erradication of S. scletoriorum is the formation of sclerotia. Why the authors did not test Xcn1 to prevent sclerotia formation or sclerotia germination? Those experiments could be most valuable to justify the use of Xcn1 as control strategy for S. sclerotiorum, on the basis that the authors showed the effect of Xcn1 on the hypha structure. I encourage the authors to performe such experiments to increase the value of the paper, and it would be more significant to the research community.

The authors could show the images from the in vitro inhibition assay, the results are interesting and the photopgraphs may complete the table 1 information. The images can go as supplementary materials.

Comments on the Quality of English Language

In line 49, the sentence "the converged mycelia and was survived in the soil" should be "the converged mycelia and survived in the soil" or "the converged mycelia and can survived in the soil".

In line 105 the sentence "were shown in Fig. 1 and Fig. 2," should be "are shown in Fig. 1 and Fig. 2,".

Reviewer 2 Report

Comments and Suggestions for Authors

Dear authors,

The manuscript is well written and has interesting data.

I would suggest to cite other manuscripts which already showed that X. nematophila has antifungal activity including the used fungal pathogens in this manuscript.

For example:

11)     Chen et al. 1994 Antifungal activity of two Xenorhabdus Species…. Biological control 4, 157-162

22)     McInerney et al., 1991 Biologically active metabolites from Xenorhabdus ssp. Part 2….. J. Nat. Prod. 54, 785-795

33)     Or the manuscript by the same authors: Guo et al., 2017

And others…..

There are also several publications describing protein biosynthesis as the mode of action for Xcn1, which should at least be discussed in the context of the morphological and physiological results shown in this manuscript:

11)     Zumbrunn C, Krüsi D, Stamm C, Caspers P, Ritz D, Rueedi G. Synthesis and Structure-Activity Relationship of Xenocoumacin 1 and Analogues as Inhibitors of Ribosomal Protein Synthesis. ChemMedChem. 2021 Mar 3;16(5):891-897. doi: 10.1002/cmdc.202000793. Epub 2020 Dec 11. PMID: 33236408.

In figure legends please include which statistics were used.

Line 183 limpid? Is it liquid?

Line 195 ideal? Is it idear?

Line 390 is it really 2.7?

Comments on the Quality of English Language

very minor changes needed. 

Reviewer 3 Report

Comments and Suggestions for Authors

This manuscript presents the results of studies documenting the strong antifungal activity of Xenocoumacin 1 (Xcn1), produced by the entomophagous nematode Xenorhabdus nematophila. There are already many studies on the antagonistic properties of this substance against fungi and Oomycota. In this study, the authors focused mainly on the possibility of biocontrol of Sclerotinia sclerotiorum. Numerous, multifaceted, well-documented experiments were carried out covering previously unexplored aspects of the antifungal effect of Xcn1. This is a manuscript containing very valuable results for science and agricultural practice. Therefore it should be published in JoF. This manuscript, however, contains numerous but minor errors or omissions that need to be corrected (see Remarks).

Remarks

The title needs correction, it is unclear. Besides, there should be a comma after 1?

Line 26 ‘species’ – should be deleted, this word is redundant in this case

Line 18 this sentence has an incorrect structure and requires correction

Line 20 this sentence contains errors: hyphae is a plural, so it should be were and not was. Besides, it should be mitochondria instead of mitochondrial

Line 37 'a wider antifungal spectrum' - this is a repetition of the text from line 28

Line 45 'rape[1]' throughout the manuscript, a space should be added before the literature number 'rape [1]' (this is to be decided by the Editorial Board)

Line 46 'Lens culinaris' is a Latin name so it should be italic. Since all other plant names are in English, this one should also be in English (to standardize the text)

Line 48 ‘sclerotia which is’ - rather ‘sclerotia which are’ (Plural).

Line 49 ‘sclerotia is a structure’ – either ‘sclerotium is’ or ‘sclerotia are..’ This should be corrected throughout the manuscript

Line 74 (Supplementary Material) – either cross out or cite a specific table

Line 79 Polygalacturonase – this should be in lower case

Table 1 it should be Alternaria alternata instead of Alternaria alternate

Table 2 S.sclerotiorum – add a space after S.

Line 211 ‘phytophthora’ – write with a capital letter

Line 222 'and infection' it is unclear, the text needs to be clarified. It was not directly examined in this work, you cited the paper [29-31].

Line 230 – 231 – this is text typical for Discussion, it should be taken out of 'Results'

Line 242-245 consider revising this sentence

Line 256 it should be Verticillium dahliae instead of Verticillium dahlia

Line 256 it should be Botrytis cinerea (italic) instead of Botrytis cinereal

Line 257 it should be Alternaria alternata instead of Alternaria alternate

Line 257 be sure to put a comma after 'gloeosporioides'

Line 257 it should be Gaeumannomyces graminis instead of Gaeumannomyces gramini

Line 356 and line 367 ‘this agar plugs of S. sclerotiorum isolates’ – were these ten different isolates or one isolate replicated ten times?

Line 415 ‘physiological and biochemical processes’ – state the most important ones as the result of current experiments

Figure 1A and Figure 2A  what does 'prentive efficacy' mean

Figure 4A should explain what does CY mean

Line 417 Supplementary Materials: - no data available

Line 472 nematophilus – should it be nematophila?

Comments on the Quality of English Language

see Remarks

Round 2

Reviewer 1 Report

Comments and Suggestions for Authors

I thank the authors for the considerations to my comments.

Just as a recomendation for future research, the authors mentioned that they did not take photographs of the in vitro experiments, you should always take photos as evidence and future references of the experiments, even if you will not use them in research papers.

I hope to see the experiments with sclerotia in future research, that will be very interesting.